# Polycaprolactone-Based Films Incorporated with Birch Tar—Thermal, Physicochemical, Antibacterial, and Biodegradable Properties

**DOI:** 10.3390/foods12234244

**Published:** 2023-11-24

**Authors:** Agnieszka Richert, Ewa Olewnik-Kruszkowska, Rafał Malinowski, Agnieszka Kalwasińska, Maria Swiontek Brzezinska

**Affiliations:** 1Department of Genetics, Faculty of Biological and Veterinary Sciences, Nicolaus Copernicus University in Toruń, Lwowska 1 Street, 87-100 Toruń, Poland; 2Department of Physical Chemistry and Physicochemistry of Polymers, Faculty of Chemistry, Nicolaus Copernicus University in Toruń, Gagarin 7 Street, 87-100 Toruń, Poland; olewnik@umk.pl; 3Łukasiewicz Research Network—Institute for Engineering of Polymer Materials and Dyes, 55 M. Skłodowska-Curie Street, 87-100 Toruń, Poland; rafalmalinowski@gmail.com; 4Department of Environmental Microbiology and Biotechnology, Faculty of Biology and Veterinary Science, Nicolaus Copernicus University in Toruń, Lwowska 1, 87-100 Toruń, Poland; kala@umk.pl

**Keywords:** polycaprolactone, birch tar, antibacterial properties, packaging materials, functional properties

## Abstract

We present new polymer materials consisting of polycaprolactone (PCL), polyethylene glycol (PEG), and birch tar (D). PEG was introduced into the polymer matrix in order to obtain a plasticizing effect, while tar was added to obtain antibacterial properties and to change the physicochemical properties of the film. The materials were obtained by the solvent method and characterized using a variety of methods to test their performance and susceptibility to biodegradation. The obtained data indicate that the introduction of the bioactive substance (D) into PCL improved the thermal stability and significantly lowered the Young’s modulus values of the tested polymers. Moreover, the addition of birch tar improved the barrier and bacteriostatic properties, resulting in a reduction in the growth of pathogenic bacteria on the surface of the film. The films are not mutagenic but are susceptible to biodegradation in various environments. Due to their properties, they have potential for application in agriculture and horticulture and for packaging food, mainly vegetables grown in the field.

## 1. Introduction

Polycaprolactone (PCL) is a biodegradable polymer belonging to the group of aliphatic polyesters. It is obtained from caprolactone by ring-opening polymerization [1]. This polymer has a valuable feature: it is easily mixed with many other polymers and is used as a plasticizer to increase the elasticity of plastics. Therefore, the use of poly(ε-caprolactone) is quite widespread. From the classic one, as a raw material for obtaining appropriate products in the processes of injection, extrusion, and extrusion blow molding, through obtaining fibers and threads, to the application as a compatibilizer in mixtures. It is commonly combined with starch to create biodegradable plastics used for producing disposable plates and cups that can be composted after use [2]. Due to its biodegradable properties, poly(ε-caprolactone) is an interesting material for many modern applications [3,4,5,6].

Tar is a substance in the form of a liquid, yellow–brown–black in color. It is a product of the distillation of the wood or bark of some trees without air at a temperature of 400–1000 °C. The most common tars are birch, pine, beech, and juniper tar, but also larch, fir, willow, and peat tar. Tar is a multi-component substance [7,8,9,10,11,12,13,14]. The most important chemical components are primarily phenol derivatives (guaiacol, creosote, pyrocatechin), betulin, benzene, xylene, phytoncides, organic acids, and resin substances [10,11,12,13,14]. Purified and distilled tar in the form of ointments and alcoholic solutions can be utilized due to its antibacterial and antifungal properties [15,16,17]. Birch tar is also an effective remedy for protecting plants against insects, rodents, snails, and weeds [7,8,15]. The existing literature primarily focuses on PCL’s applications in the medical field, such as hydrogel dressings, implants, or surgical threads.

Due to the need for bactericidal materials to appear on the market, it was checked whether it would be possible to combine tar and biodegradable polymer material [10,15,16,17]. Having obtained promising research results, it was decided to perform analyses on another biodegradable material as well, i.e., polycaprolactone.

The aim of the research was to determine the properties of polycaprolactone modified with birch tar for food packaging applications of produce such as vegetables and fruits. The work included performing FTIR-ATR spectra, TG analysis, mechanical tests, assessment of color and transparency, water vapor and oxygen permeability, bactericidal and mutagenic assessment, and analysis of biodegradation in compost, soil, river, and lake water.

## 2. Materials and Methods

### 2.1. Materials

Poly(caprolactone) (PCL), type CAPA 6800 (Solvay Caprolactones, Warrington, UK). Chloroform, methanol, and polyethylene glycol (PEG) (Avantor Performance Materials Poland S.A., Gliwice, Poland). Birch tar (*Betula pendula*) Gift of Nature (Gift of Nature, Grodzisk, Poland) and was used as an antibacterial agent.

Bacterial reference strains of *Escherichia coli* (ATCC 8739), *Staphylococcus aureus* (ATCC 6538P), and *P. aeruginosa* (ATCC 35554) (Argenta, Poznań, Poland) and environmental strains of *A. tumefaciens*, *X. campestris*, *P. brassicacearum*, *P. corrugate*, and *P. syringae* (Department of Environmental Microbiology and Biotechnology, Faculty of Biology and Veterinary Science, NCU, Toruń, Poland) were incubated in tryptone peptone—15 [g/L], phyton peptone—5 [g/L], and sodium chloride—5 [g/L], and were delivered by Oxoid (Oxoid, Hampshire, UK).

Compost, soil (biodegradation environment, Rozgarty, Poland, DMS: 53°26′20.192″ N 18°41′11.693″ E), water river (Vistula River, Toruń, Poland, DMS: 53°0′52.33″ N 18°35′47.699″ E), Lake Chełmżyńskie, Poland, DMS: 53°11′32.553″ N 18°36′45.196″ E).

### 2.2. Polycaprolactone-Based Materials Formation

Polycaprolactone-based films were prepared in the procedure using the solvent method, described in [10,15,18,19,20]. The examined films were prepared using a laboratory method. Polycaprolactone pellets were dissolved in chloroform in an attempt to obtain a 3% (*w*/*v*) polymer solution. Subsequently, 1, 5, or 10% wt. of D (in the form of an oily solid) was added to the PCL/PEG solution. In total, 5% wt. PEG was introduced into the solutions to prepare plasticized PCL films. To obtain plasticized PCL materials, 50 mL of the prepared mixture was poured onto glass Petri dishes (14.5 cm in diameter) and left for 3 days to form a polymer film. The control film (marked with C) consisted of PCL plasticized with PEG. The samples (Figure 1), in addition to the aforementioned components, also included 1%, 5%, and 10% by mass of birch. The prepared materials, together with the increasing tar content, were marked as C, CD1, CD5, and CD10, respectively.

### 2.3. FTIR Analysis

The study of the impact of the introduction of birch tar on the structure of the obtained films was conducted using a Nicolet iS10 spectrometer (Thermo Fisher Scientific, Waltham, MA, USA).

The frequency range of 500–4000 cm^−1^ and a resolution of 4 cm^−1^ were applied in all 64 scans during the recording of the spectra. With the aim of analyzing the obtained data, the OMNIC 7.0 software (Thermo Fisher Scientific, Waltham, MA, USA) was applied.

### 2.4. Thermal Properties

The thermal stability of the obtained materials was tested using the thermogravimetric method. The SDT 650 thermal analyzer from TA Instruments was used for analysis. The analysis was performed in an air atmosphere at a heating rate of 10 °C/min. The analysis was performed in the temperature range of 20 °C to 600 °C.

### 2.5. Mechanical Properties

The mechanical properties such as elongation at break (σ_M_), tensile stress (m), and Young’s modulus (E) were studied using the Shimadzu EZ-test SX machine (Kyoto, Japan). The TRAPEZIUMX (C224-E048A) software was used to analyze the results, and the testing followed the ISO standards [21,22].

### 2.6. Color Analysis

In order to determine the color change of the obtained materials after adding tar, a DR LANGE MICRO COLOR II GB colorimeter was used. A white standard (X = 75.2; Y = 80.1; Z = 83.9) was used to calibrate the device. Each sample was measured in three places, and the average value was calculated.

The total color difference (ΔE) and color intensity (C) were calculated using Formulas (1) and (2) below:(1)$$\Delta E=\sqrt{{\left(L-L*\right)}^{2}+{\left(a-a*\right)}^{2}+{\left(b-b*\right)}^{2}}$$
(2)$$C=\sqrt{{\left(a\right)}^{2}+{\left(b\right)}^{2}}$$
where L*, a*, and b* represent standard values for samples without any additive (sample C), and L, a, and b are the average values of the measurements for each film tested. Additionally, the yellowness index (YI) was determined using Equation (3):(3)$$YI=\frac{142.86\cdot b}{L}$$

### 2.7. Transparency

The transparency of the studied materials was measured at 600 nm (A600) using the UV spectrophotometer Halo DB-20 (Dynamica Scientific Ltd., Newport Pagnell, UK, Great Britain) according to the method described previously [23]. The transparency (T) of the polymeric films was calculated using Equation (3):(4)$$T\left[{mm}^{-1}\right]=A_{600}÷d$$
where d represents the film thickness [mm].

### 2.8. Permeability of Water and Oxygen Transmission Rate (OTR)

The water vapor permeability of the tested materials was assessed as follows: Approximately 5 g of dried calcium chloride was placed in a container (25 mm in diameter). The film to be tested was cut out and placed on the top of the prepared container. The container was sealed with a tape. The weight of the calcium chloride container was recorded before placing the sample in the incubator (30 °C, 75% relative humidity). The sample was weighed every 24 h for 7 days. Two replicates were performed for each sample. In order to determine the WVTR (water vapor transmission rate), weight gains over time were plotted, and then the WVTR value was calculated using formula No. 1:(5)$$WVTR\left[\frac{{\frac{g}{m}}^{2}}{day}\right]=\frac{the\;slope\;of\;the\;straight\;line}{film\;surface}$$

The value obtained from Equation (1) was additionally multiplied by the thickness of the tested film.

The oxygen transmission rate (OTR) of the polymer films was determined by measuring the changes in oxygen content in the distilled water as a recipient using Winkler’s method [23]. Deionized water was boiled for 15 min. to remove dissolved oxygen, and then 50 mL was transferred to a plastic container (40 mm in diameter) and finally covered with polymer films. The open container, allowing oxygen to enter the flask and dissolve in the water freely, was used as a control. The flasks were placed in an open environment at room temperature for 24 h. The results were expressed as the amount of dissolved oxygen (DO). All the studies were carried out in triplicate. The oxygen transmission rate (OTR) was then calculated:$$OTR\left[g\cdot m^{-2}\cdot h^{-1}\right]=DO\cdot \frac{V}{A\cdot t},$$
where DO—is the dissolved oxygen [g dm^3^], V—the volume of the water used [dm^3^], t—time [h], and A—the area of the film exposed for oxygen permeation [m^2^].

### 2.9. Antibacterial Effect

The antibacterial properties of the films were determined according to the ISO standard (ISO 22196:2011) using the following bacterial strains that are plant pathogens: *A. tumefaciens*, *X. campestris*, *P. corrugata*, *P. brassicacearum*, and *P. syringae*; and human pathogens: *S. aureus*, *E. coli*, *P. aeruginosa*, and *Salmonella.* Antibacterial activity (R) was determined according to the guidelines specified in the standard [24].

### 2.10. Ames Test

A mutagenicity test was performed to determine the direct effect of the test substance or its metabolites on the cell’s genotype. The potential mutagenicity of the test samples was analyzed using the in vitro method according to the Ames test. M9 minimal medium (Na_2_HPO_4_ × 12 H_2_O, KH_2_PO_4_, NaCl, NH_4_Cl, agar, MgSO_4_, CaCl_2_, glucose, ampicillin, biotin, histidine) was inoculated with a Salmonella typhimurium strain. Then, the samples were placed on the prepared pans. The controls constituted Petri dishes containing only the medium with the Salmonella typhimurium-inoculated strain. All the plates were incubated upside down and wrapped in aluminum foil for two days at 37 °C. The lack of growth of a significant number of bacterial cells around the sample indicates that the film is not mutagenic.

### 2.11. Biodegradability

The biodegradability of the films was analyzed in four environments: fertile environments (soil and compost) and less fertile environments (lake water and river water).

Biological oxygen consumption in river water and lake water was determined using the method described by Swiontek Brzezinska et al. (2021) [25]. The procedure involved filling a 0.5-L bottle with 430 mL of lake/river water, placing a magnetic stirrer inside, adding five drops of the nitrification inhibitor NT 600 (WTW, Wrocław, Poland), inserting three film samples sized 5 cm × 5 cm, and including a rubber quiver containing ~0.4 g of NaOH as a CO_2_ absorbent. Subsequently, OxiTop heads were securely attached to the prepared cylinders, and the BOD measurements in triplicate were initiated and expressed in mgO_2_/L.

The determination of biodegradability in compost and soil was carried out using the method described by Platen and Wirtz (1999) [26]. One hundred grams of soil were placed in a one-liter jar and mixed with three film samples sized 5 cm × 5 cm. The rubber quiver contained ~2.5 g of NaOH. Soil/compost respiration measurements were initiated using the pressure mode and expressed in mgO_2_/kg of soil/compost after 14 days at 20 °C. All samples were analyzed in triplicate.

## 3. Results and Discussion

### 3.1. Fourier Transform Infrared Spectroscopy

The FTIR spectrum is an effective technique for studying the interactions between functional groups based on the shifts of the vibrational bands. The increase in band intensity at 758 cm^−1^–757 cm^−1^ indicates that birch tar affects the crystallinity of the obtained materials. In Figure 2, the infrared absorption spectrum of poly(ε-caprolactone) with a 5% *w*/*w* addition of poly(ethylene glycol) is presented, while Table 1 shows the characteristic bands for pure PCL with the obtained results.

According to Coleman and Zarian [27], the bands at 1294 cm^−1^ and 1167 cm^−1^ are assigned to the crystalline and amorphous forms of PCL, respectively. In the case of sample C, the intensity of the band in the range of 1301 cm^−1^–1249 cm^−1^ is almost twice as high as in the range of 1162 cm^−1^–1156 cm^−1^, which can lead to the conclusion that the crystalline form prevails in the tested material.

The figure below shows the FTIR spectra for all the tested samples based on PCL.

As presented in the work by Lyu et al. [28], in the FTIR spectra of poly(ε-caprolactone) with the addition of birch tar, no significant changes in the intensity of the bands are observed with increasing concentrations of the additive in the PCL matrix. In the cited publication, grapefruit seed extract (GSE) was added to the PCL matrix. The bands in the pure PCL film spectrum were similar to the PCL/GSE composite bands. The only difference between the sample without the addition of GSE and the sample with the addition of GSE was the presence of a new band at 3300 cm^−1^, which was attributed to the typical vibrations of the -OH group of phenolic/aromatic compounds in the grapefruit seed extract. In the case of the tested sample marked as C, the band at 3437 cm^−1^ was visible in the FTIR spectrum due to -OH groups at the ends of PCL chains and -OH groups included in the poly(ethylene glycol). This is due to the presence of tar compounds such as phenol, guaiacol, cresol, pyrocatechin, and other aromatic compounds. The lack of new bands between the sample without the addition of tar and the sample with the addition of tar suggests that there were no significant structural changes caused by the addition of birch tar to the PCL matrix, such as the formation of new chemical bonds. The authors of the work quoted above came to the same conclusions.

### 3.2. Thermogravimetric Analysis

In order to determine the effect of different amounts of birch tar on the thermal stability of the film based on poly(ε-caprolactone), a thermogravimetric analysis was carried out. The following figures show the thermogravimetric curve (Figure 3a) and its derivative (Figure 3b) of materials based on poly(ε-caprolactone) with the addition of birch tar.

According to the literature, the temperature at which pure PCL begins to decompose is 382 °C [29]. On the TG curve (Figure 3), it is clearly visible that this temperature is much lower and amounts to about 284 °C. In the work by Tao Sun and Xiaomin Shuai [4], where PCL-PEG-PCL copolymers were tested, it was shown that such a copolymer is stable up to 267 °C. This value is close to the one obtained in this work. The difference in the decomposition temperatures of pure PCL and the tested sample C may therefore be due to the presence of a 5% *w*/*w* addition of poly(ethylene glycol), as in the case of PLA-PEG materials.

In order to determine how different concentrations of birch tar, which was added to the PCL film as a fungicide, affect the thermal stability of the obtained material, the temperatures at which the mass loss occurs were determined. Below is a graph (Figure 3) showing the temperature at 10%, 30%, and 50% weight loss.

The presented results indicate that the addition of birch tar to the poly(ε-caprolactone) matrix has a positive effect on the thermal stability of the tested films (Figure 4). In all the decomposition stages tested, the thermal stability increases with the concentration of the active ingredient; therefore, the highest decomposition temperature is observed for the CD10 sample. The greatest differences in temperature between samples with different concentrations of tar are observed in the initial stage of the decomposition (10% weight loss). The difference between the control sample and the sample with 10% *w*/*w* tar is 37.4 °C.

The increasing thermal stability of PCL materials was also noted by Silva et al. [3] for materials containing andiroba oil. The control sample, which did not contain the oil, decomposed in the temperature range of 242.7–450 °C, which coincides with the results of the present work (Figure 3). The incorporation of andiroba oil into the poly(ε-caprolactone) matrix resulted in shifting the decomposition temperatures to higher values. This increase in degradation temperatures may be related to the barrier effect caused by the polymer chains against the oil molecules, which impedes the diffusion of volatile compounds and thus increases the thermal stability of the tested materials. Similar observations were also presented in the work by Reshmi et al. [30], where membranes made of PCL and beeswax were tested. The increase in degradation temperatures was attributed to the addition of wax, the presence of which caused a barrier effect, hindering the diffusion of degradation products from the polymer mass.

### 3.3. Film Thickness

The resulting films were measured at 20 different locations with an accuracy of ±0.001 mm. Table 2 summarizes the obtained results, showing means and standard deviations.

The thickness does not depend on the added tar, and changes in the thickness of the film are small. The obtained results show similarity with the results obtained earlier for the same type of film [20].

### 3.4. Changes in Mechanical Properties

With the aim of establishing the effect birch tar has on the mechanical properties of the studied materials, the Young’s modulus, tensile strength, and elongation at break were studied and discussed. Young’s modulus is a coefficient of proportionality between the stress and the relative elongation of the material [31]. The higher the value of Young’s modulus, the harder, stiffer, and less flexible the material is. In the case of the studied materials, the values of the Young’s modulus of samples with the addition of birch tar were reduced compared to the control sample without the biocidal substance (Figure 5a). Fortunately, the reduction in Young’s modulus is not significant. The lowest value of the elastic modulus was observed for the film containing 10% birch tar, and it was approximately 50 MPa lower than in the case of the material without the addition of birch tar. It should be noted that birch tar was introduced in the form of wax, which may play a role similar to a plasticizer. For this reason, a decrease in the Young’s modulus was expected. Moreover, it should be emphasized that the incorporation of an additive into the polymer matrix, in most cases, leads to an increase in the free volume between the polymer chains and is one of the other factors causing a decrease in the Young’s modulus value.

The elongation at break is another parameter taken into account when specifying the properties of packaging materials. It is the ratio of the change in the length of the sample at the moment of rupture to the initial length of the sample [32]. The obtained results clearly indicate that the addition of birch tar increased the flexibility of the PCL-based foil. The most significant difference compared to the control film can be observed in the case of sample CD10, where the elongation at break was 54.7% while for the control sample it was 35.6% (Figure 5b). The obtained results are in accordance with the hypothesis mentioned above that birch tar acts as a plasticizer.

In the majority of polymeric materials, tensile strength is an additional parameter describing mechanical properties. Tensile strength is the stress corresponding to the highest tensile force obtained during a static tensile test, related to the original cross-sectional area of the analyzed sample. The obtained results allow us to establish that the incorporation of birch tar does not significantly influence the values of tensile strength (Figure 5c). This proves that the force needed to rupture the material remains almost unchanged. Only in the case of the sample containing 5% birch tar was a slight increase in the tensile strength observed. However, the introduction of a higher amount of the additive into the polymer matrix causes the parameter to return to the previous level.

The obtained results lead to the conclusion that the introduction of birch tar into the analyzed polymer matrix reduces the stiffness of the material, significantly improves its elasticity, and, at the same time, does not affect the strength of the obtained foils.

### 3.5. Analysis of the Color and Transparency of the Film

Color is an important characteristic of the packaging material. The color of the packaging can create the most appropriate impression when the consumer interacts with a given product; therefore, manufacturers modify the color of the packaging in order to distinguish their goods from the competition [33].

The color of the packaging material can not only improve its aesthetics but also affect its functional properties. For example, the dark color of the packaging (brown or green) may guarantee protection of the product against degradation caused by UV radiation [34]. The values of the measured parameters characterizing the color of the films (L, a, and b) are listed in Table 3.

The addition of birch tar makes the foil darker. The greater the addition of birch tar, the more red and yellow hues the foil acquires (Table 3).

Total color difference (ΔE) and color intensity (C) are shown in the Figure 6a–c.

A human cannot perceive a color difference if ΔE is less than 3 [35]. In the case of the tested films, the color difference was clearly visible to the naked eye, which is also confirmed by the obtained values of the ΔE parameter.

The change in the color of the film was not intentional and was caused by the addition of brown–red birch tar. The addition of birch tar also affects the opacity of the obtained films. There is a visible tendency to increase the haze of the film with increasing concentrations of birch tar. Comparing the sample without the addition of tar and the sample with the highest addition of tar, an almost sevenfold increase in the turbidity of the material was observed.

In the work by Pritchard [36], it was noticed that the lower the density of a crystalline polymer, the more transparent the polymer is. This parameter is related to the degree of crystallinity of the polymer—a material with a greater degree of crystallinity has a higher density. The obtained results may suggest that polymer films containing birch tar can be used as packaging materials, especially for products sensitive to light, due to their turbidity and dark color.

### 3.6. Water Vapor Permeability and OTR

Determining the water vapor transmission rate (WVTR) is an important element when assessing the barrier properties of the film. Reducing the migration of water vapor from the external environment to the product, or vice versa, reduces the unfavorable changes in the quality of the product and increases its shelf life [37].

In the case of PCL, the impact of tar on the film’s water vapor permeability was prominently evident (Figure 7). A mere 1% addition of tar notably enhanced the material’s barrier properties, making it the best-performing among the tested films. However, further increases in the concentration of birch tar did not yield a positive effect on the material’s permeability. The difference in water vapor permeability between sample C and sample CD10 was 42%.

The OTR results are shown in Figure 8. For samples containing tar (CD1, CD5, CD10), OTR was lower than for the control sample C by 8.7, 18.5, and 28.3%, respectively.

Penetration of oxygen from the environment into packaged food has a significant impact on the quality and durability of food. OTR packaging is very important due to the effect of oxygen on food in packages, including fruit and vegetables. Oxygen causes food to spoil through the oxidation of lipids and vitamins, leading to sensory disorders and changes in nutrient composition. The addition of tar to PCL increased the polarity or hydrophilicity of the composite, which makes it a better barrier to nonpolar materials such as oxygen.

### 3.7. Antibacterial Properties

The biocidal properties of the birch tar containing PCL films are presented in Table 4.

The results showed antibacterial properties of the films against all tested pathogens.

In our previous studies, we determined the biocidal properties of tar films with PLA as the polymer matrix. Birch tar is an effective biocidal substance that has not been noticed before [38].

The antimicrobial properties of birch tar are due to the presence of phenolic compounds. Phenols seem to be one of the most promising groups of compounds in BTO acting as biocontrol agents, but various volatile compounds may also play a role [7]. Among the phenolic compounds, cresols, allylphenol, guaiacol, 4-methyl- and 4-ethyl guaiacol, eugenol, isoeugenol, vanillin, and ethylvanillin have been identified in birch biomass pyrolysis [10,11,12,13,14].

The antimicrobial mechanism of action of polyphenols is different. It may be related to disruption of enzymatic processes, e.g., dehydrogenases involved in cellular respiration. Polyphenols, e.g., eugenol or vanillin, have antioxidant properties [39].

Polyphenols also hinder the formation of microbial biofilms [40,41,42,43]. It is possible that the birch tar present in the tested polymers, containing numerous polyphenols, is responsible for the biocidal effect.

### 3.8. Ames Test—Mutagenicity Method

The Figure 9 shows test results, providing information on the mutagenicity of polymer films. The results obtained indicate the lack of mutagenicity of the tested samples: C, CD1, CD2, and CD3. The results showed that tar and PEG added to polycaprolactone do not create a biopolymer with mutagenic properties. This is a positive discovery because tar contains a number of substances that are mutagenic or toxic; however, when combined with the polymer, these properties disappear.

### 3.9. Biodegradability

Biological oxygen consumption by microorganisms in the presence of the tested films is presented in Table 5.

The obtained data indicate that soil and compost are the best environments for the biological degradation of PCL materials with tar. The values of BOD obtained for all samples containing tar in soil and compost are orders of magnitude higher than those obtained for aqueous environments.

It is true that each of the test samples had lower oxygen consumption values than the control samples, which means that birch tar actively hampers microorganisms’ growth. Similar results were obtained for PLA foils containing tar [2,3,20].

## 4. Conclusions

The presence of birch tar in the plasticized PCL matrix has a significant impact on the properties of the resulting films. It notably enhances the barrier properties compared to samples without the additive, increases material elasticity, imparts a dark color, and reduces transparency. This reduction in transparency can be beneficial for packaging, as it provides protection against UV rays. Plasticized PCL materials with the addition of birch tar show promise as potential packaging materials, particularly with regard to flexible packaging. Therefore, using this polymer for packaging is an innovative and relatively unexplored area. These materials appear to have great potential for use in horticulture and agriculture packaging.

## Figures and Tables

**Figure 1 foods-12-04244-f001:**
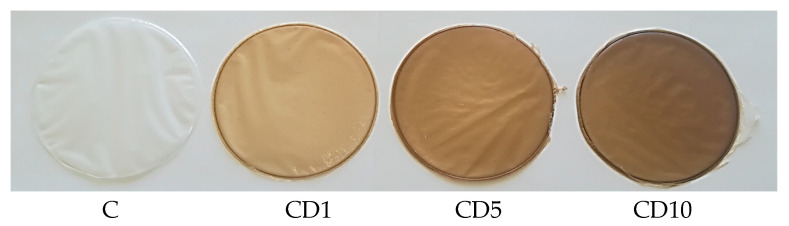
Macroscopic appearance of the films.

**Figure 2 foods-12-04244-f002:**
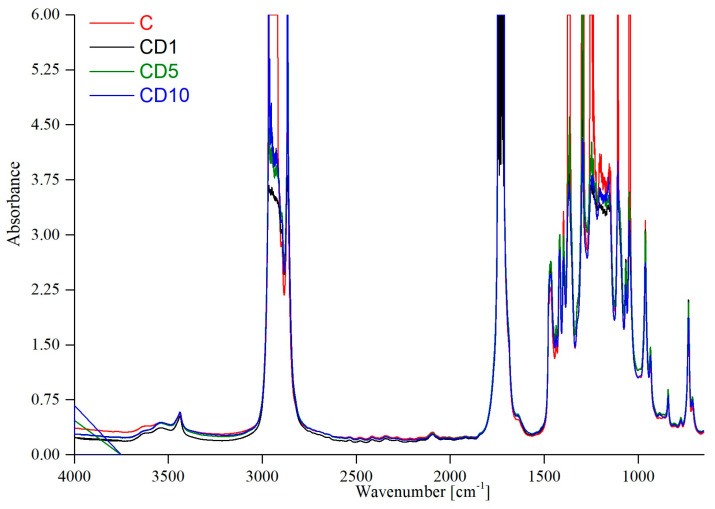
FTIR spectra of samples based on PCL with the addition of birch tar.

**Figure 3 foods-12-04244-f003:**
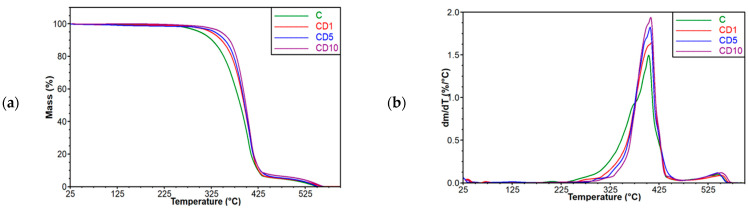
TG curve: (**a**) TG curve for PCL-plasticized materials; (**b**) derivative of the TG curve for PCL-plasticized materials.

**Figure 4 foods-12-04244-f004:**
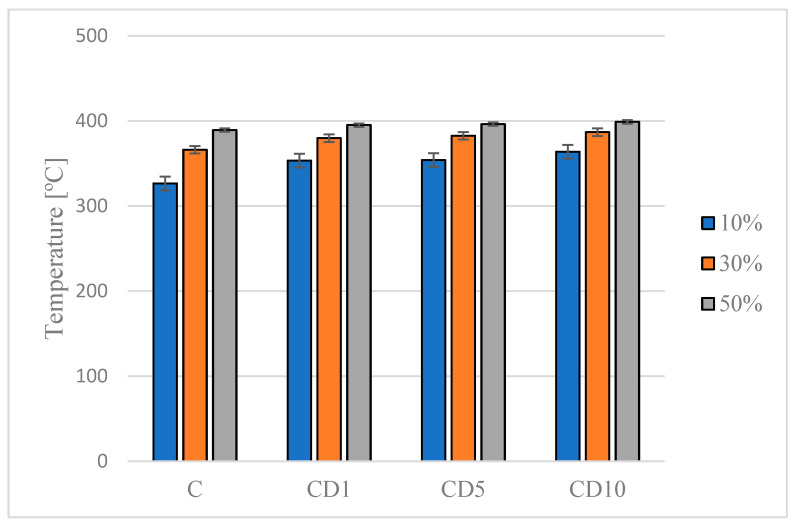
The weight loss of the PCL-based materials.

**Figure 5 foods-12-04244-f005:**
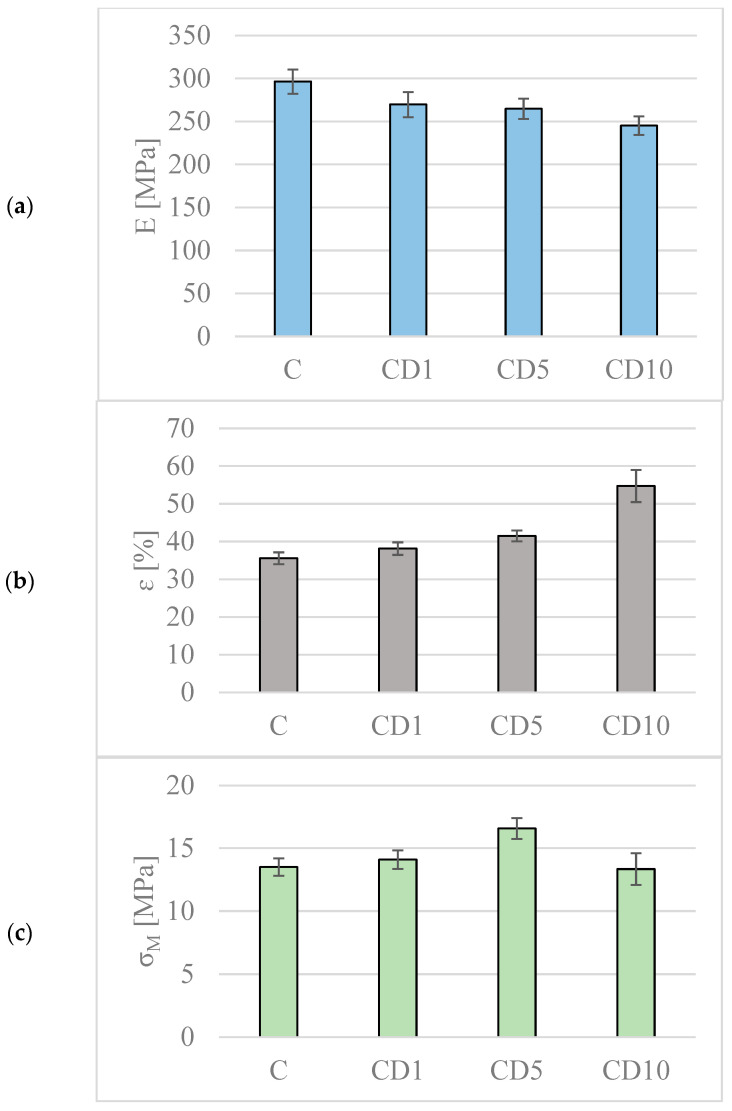
Values of the mechanical properties of obtained films: (**a**) Young’s modulus (E); (**b**) elongation at break (ε); (**c**) tensile strength (σ_M_). Different colors were used for different parameters.

**Figure 6 foods-12-04244-f006:**
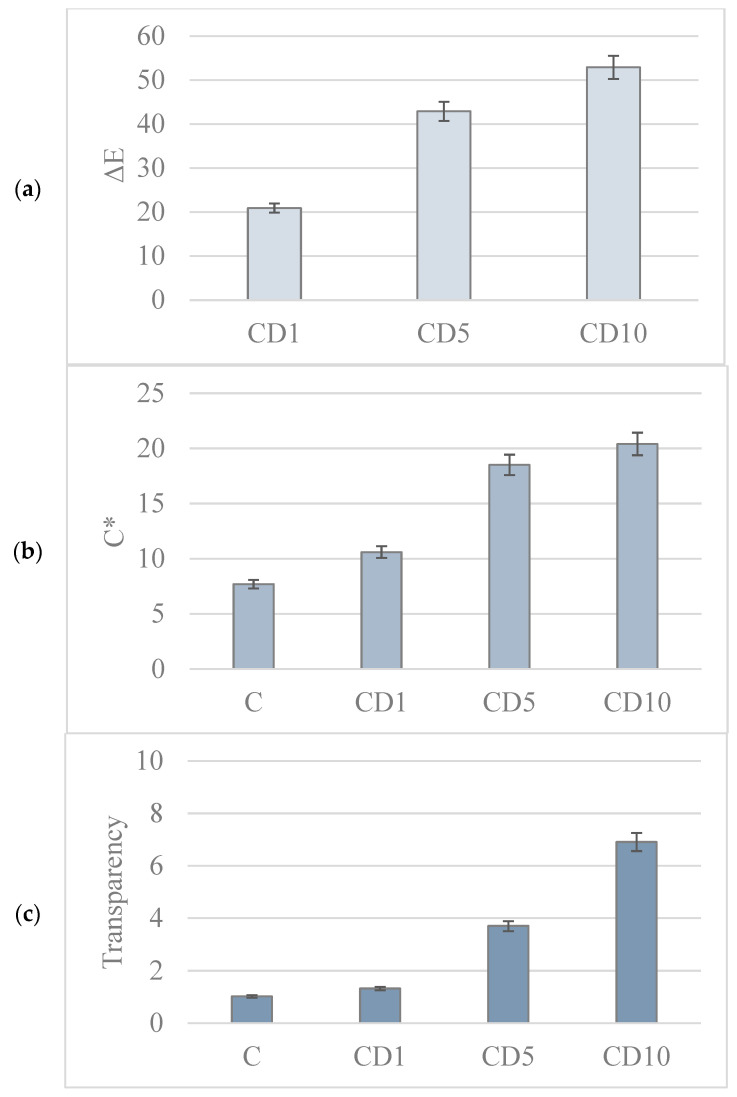
Color parameters: (**a**) ΔE; (**b**) YI; (**c**) color intensity; (**d**) transparency of studied polymeric films. Different colors were used for different parameters.

**Figure 7 foods-12-04244-f007:**
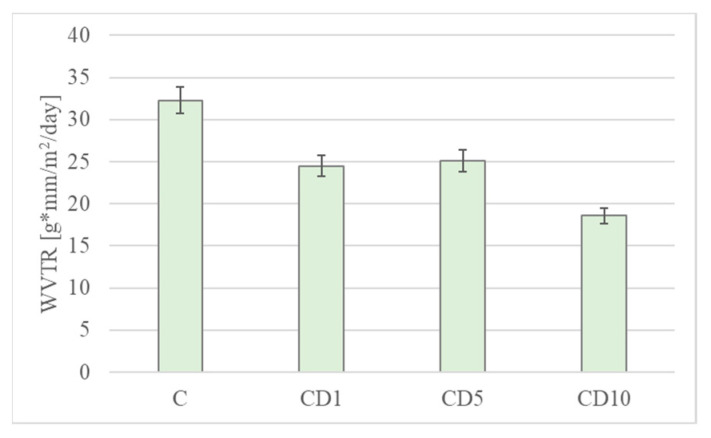
Permeability of water vapor coefficients of a film based on PCL with the addition of birch tar. In the case of PCL, the reduction in the water vapor transmission coefficient, and consequently the enhancement of barrier properties, is attributed to the hydrophobic nature of birch tar. Variations in the impact of birch tar on the properties of the tested materials may stem from differences in the degree of crystallinity among the polymers used. In summary, birch tar, owing to its hydrophobic properties, can serve as an additive to polymers to reduce their water vapor transmission coefficient and, consequently, enhance the packaging’s barrier properties.

**Figure 8 foods-12-04244-f008:**
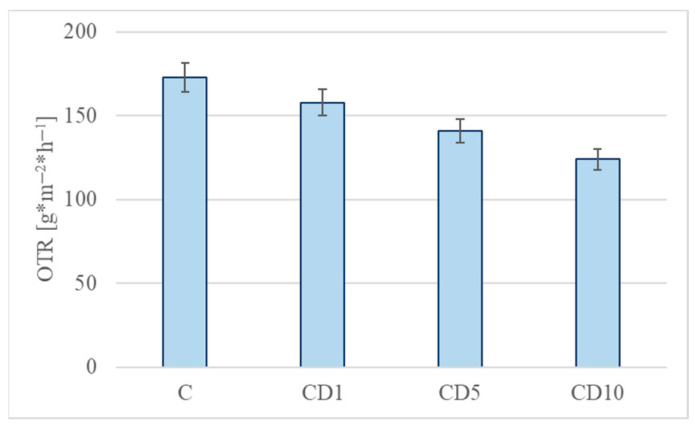
Oxygen transmission rate (OTR) films: C, CD1, CD5, and CD10.

**Figure 9 foods-12-04244-f009:**
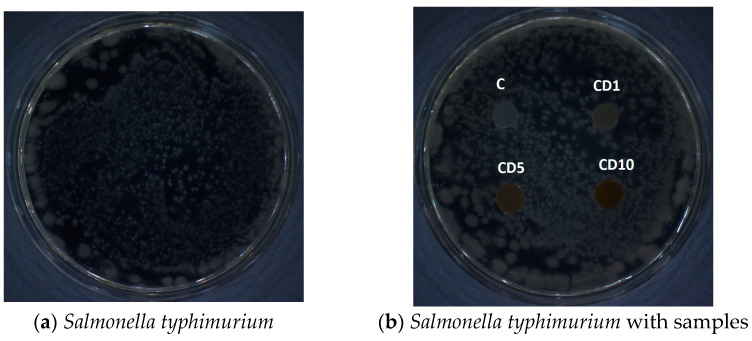
Ames test: (**a**) control—a culture of *Salmonella typhimurium* (reverse control); (**b**) tested samples in the presence of the *Salmonella typhimurium* strain (induction of mutation with the tested polymer materials).

**Table 1 foods-12-04244-t001:** Comparison of the FTIR bands of the obtained PCL sample with the literature.

Description	Clear PCL [Lyu, Khatiwala] [cm^−1^]	Sample C [cm^−1^]
Symmetric and asymmetric stretching vibrations CH_2_	2943 2864	2918 2864
Stretching vibrations C=O, carbonyl group	1727	1726
Stretching vibrations C-C	1294	1249 1301
Stretching vibrations C-O-C	1167	1156 1158 1162

**Table 2 foods-12-04244-t002:** Thickness of the obtained materials based on plasticized PCL without and with the addition of birch tar.

Samples	C	CD1	CD5	CD10
Thickness [mm]	0.088 ± 0.008	0.082 ± 0.011	0.064 ± 0.010	0.077 ± 0.010

**Table 3 foods-12-04244-t003:** Color characteristics of the films: C, CD1, CD5, and CD10.

Samples	L	a	b
C	91.3 ± 0.1	1.0 ± 0.0	−10.6 ± 0.3
CD1	79.9 ± 1.1	2.2 ± 0.0	6.9 ± 1.1
CD5	59.2 ± 1.8	6.7 ± 0.4	17.3 ± 0.5
CD10	47.8 ± 0.2	9.0 ± 0.1	18.3 ± 0.3

**Table 4 foods-12-04244-t004:** Antimicrobial activity of CCD1, CD5, and CD10 samples with reference to the control sample—C against pathogen strains.

Sample Description	Sample Description	R	% Reduction	Antimicrobial Efficacy
*X. campestris*	C	-	-	-
	CD1	2.2	>99.9	very good
	CD5	2.8	>99.9	very good
	CD10	3.2	>99.9	very good
*P. brassicacearum*	C	-	-	-
	CD1	1.9	>90.0	satisfactory
	CD5	2.1	>99.9	very good
	CD10	2.5	>99.9	very good
*P. corrugata*	C	-	-	-
	CD1	2.1	>99.9	very good
	CD5	2.4	>99.9	very good
	CD10	2.8	>99.9	very good
*P. syringae*	C	-	-	-
	CD1	1.9	>90.0	satisfactory
	CD5	2.0	>99.9	very good
	CD10	2.2	>99.9	very good
*E. coli* (ATCC 8739P)	C	-	-	-
	CD1	1.5	>90.0	satisfactory
	CD5	1.7	>99.9	satisfactory
	CD10	1.9	>90.0	very good
*S. aureus* (ATCC 65388)	C	-	-	-
	CD1	1.7	>90.0	satisfactory
	CD5	1.8	>90.0	satisfactory
	CD10	2.0	>99.9	very good
*P. aeruginosa* (ATCC 8739)	C	-	-	-
	CD1	2.0	>99.9	very good
	CD5	2.3	>99.9	very good
	CD10	2.5	>99.9	very good

**Table 5 foods-12-04244-t005:** Biological oxygen consumption in various biodegradability environments after 14 days.

Sample	BOD after 14 Days of Biodegradation in:			
	Water River	Soil	Compost	Water Lake
C	530 ± 0.12 ^d^	520 ± 0.26 ^b^	602 ± 0.15 ^b^	61 ± 0.33 ^c^
CD1	25 ± 0.09 ^c^	252 ± 0.11 ^b^	216 ± 0.10 ^b^	28 ± 0.41 ^b^
CD5	20 ± 0.23 ^d^	176 ± 0.05 ^c^	204 ± 0.21 ^a^	23 ± 0.20 ^a^
CD10	3 ± 0.15 ^d^	160 ± 0.34 ^b^	96 ± 0.31 ^b^	8 ± 0.05 ^c^

Explanations: value represents mean (*n* = 3) and standard deviation; different letters in a row indicate significant differences between means (*p* < 0.05).

## Data Availability

The data presented in this study are available on request from the corresponding author.

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
