# Peer review of "Polycaprolactone-Based Films Incorporated with Birch Tar—Thermal, Physicochemical, Antibacterial, and Biodegradable Properties"

_foods, 2023, doi:10.3390/foods12234244_

Round 1

Reviewer 1 Report

Comments and Suggestions for Authors

The authors present interesting results about new functional and biodegradable materials.

Some points should be adressed before to the paper can be accpeted for publication. 

Abstract: Should include some initial contextualization.

Introduction: What are the potential applications of the films produced? What is the objectice of producing these biodegradable materials?

Methods: The procedure used to produce the films must be stated. What was the concentration of plasticizer? 

Why the abbreviation of elongation at break is "? Please correct. (line 92)

The point 2.9. should not be called "Bactericidal properties", the authors are no evaluating the bactericial activity of the films. "Antibacterial activity" should be used instead.

The legend of the axis of the Figure 3 a) and 3b) should be corrected to english. 

Comments on the Quality of English Language

Some minor corrections should be made across the manuscript.

Author Response

Dear Reviewer,

thank You for all comments.

Best Regards

Agnieszka Richert

Reviewer 2 Report

Comments and Suggestions for Authors

Comments

In this study, an antibacterial and biodegradable PCL-based material was developed via the solution method by mixing the PEG and birch tar. The films exhibited good thermal stability and mechanical properties, good WVTR barrier performance. However, there are a large number of studies on the biodegradable packaging materials. The performance is not exciting. And deep discussion about the mechanism is seriously needed. The advantage of biodegradable packaging materials is not clear. To improve the quality of the manuscript, some suggestions are as follows:

1.     The language of the manuscript is required to be carefully revised by native English speakers.

2.     The birch tar primarily consists of phenol derivatives (guaiacol, creosote, pyrocatechin), betulin, benzene, xylene, phytoncides, organic acids and resin substances. The benzene and xylene are harmful to humans’ healthy. As an additive in the prepared PCL-based materials, the toxicity tests need to be added. Or supplement migration tests for hazardous substances during application.

3.     The introduction should be revised to include an overview of relevant research progress about PCL-based materials.

4.     Materials and Methods: Please briefly describe the preparation of PCL-based film.

5.     The abbreviations and full name should be unified, while the usage of poly(ε caprolactone), Polycaprolactone and PCL in the text is confused, as well as the poly(ethylene glycol) and PEG.

6.     What’s the meaning of “In the case of packaging intended for liquids, e.g. bottles, the material should be hard, 240 stiff and durable. On the other hand, dry products that are not exposed to damage can be 241 packed in thin, flexible foils that are easy to tear. 

The authors should declare that the prepared packaging materials are mainly intended to package what kind of food

7.     Discuss the changes in mechanical properties. 

8.     The OTR is another important element for the packaging materials. Please added.

9.     The antibacterial mechanism of the CD samples should be added.

10.  It is recommended to add packaging application cases by using the as-prepared films.

Comments on the Quality of English Language

 The language of the manuscript is required to be carefully revised by native English speakers.

Author Response

(The authors gave the same response as above.)

Round 2

Reviewer 2 Report

Comments and Suggestions for Authors

The authors have  addressed the comments raised by the editor and reviewers point by point. Based on these, acception of this manuscript is recommended.